# Effects of Long-Term Heavy Metal Exposure on the Species Diversity, Functional Diversity, and Network Structure of Oral Mycobiome

**DOI:** 10.3390/microorganisms13030622

**Published:** 2025-03-07

**Authors:** Jia Li, Shuwei Pei, Lu Feng, Jiangyun Liu, Qiwen Zheng, Xingrong Liu, Ye Ruan, Weigang Hu, Li Zhang, Jingping Niu, Tian Tian

**Affiliations:** 1School of Public Health, Lanzhou University, Lanzhou 730020, China; 220220912470@lzu.edu.cn (J.L.); peishw@lzu.edu.cn (S.P.); liujy21@lzu.edu.cn (J.L.); 220220913191@lzu.edu.cn (Q.Z.); liuxr@lzu.edu.cn (X.L.); ruany@lzu.edu.cn (Y.R.); zhangli@lzu.edu.cn (L.Z.); 2School of Stomatology, Lanzhou University, Lanzhou 730020, China; lufeng@lzu.edu.cn; 3State Key Laboratory of Herbage Improvement and Grassland Agro-Ecosystems, College of Ecology, Lanzhou University, Lanzhou 730020, China; huweigang@lzu.edu.cn

**Keywords:** heavy metal contamination, oral mycobiome, trophic mode, keystone taxa, network analysis

## Abstract

Oral fungal homeostasis is closely related to the state of human health, and its composition is influenced by various factors. At present, the effects of long-term soil heavy metal exposure on the oral fungi of local populations have not been adequately studied. In this study, we used inductively coupled plasma–mass spectrometry (ICP-MS) to detect heavy metals in agricultural soils from two areas in Gansu Province, northwestern China. ITS amplicon sequencing was used to analyze the community composition of oral buccal mucosa fungi from local village residents. Simultaneously, the functional annotation of fungi was performed using FUNGuild, and co-occurrence networks were constructed to analyze the interactions of different functional fungi. The results showed that the species diversity of the oral fungi of local populations in the soil heavy metal exposure group was lower than that of the control population. The relative abundance of *Apiotrichum* and *Cutaneotrichosporon* was higher in the exposure group than in the control group. In addition, *Cutaneotrichosporon* is an Animal Pathogen, which may lead to an increased probability of disease in the exposure group. Meanwhile, there were significant differences in the co-occurrence network structure between the two groups. The control group had a larger and more stable network than the exposure group. Eight keystone taxa were observed in the network of the control group, while none were observed in that of the exposure group. In conclusion, heavy metal exposure may increase the risk of diseases associated with *Apiotrichum* and *Cutaneotrichosporon* infection in the local populations. It can also lead to the loss of keystone taxa and the reduced stability of the oral fungal network. The above results illustrated that heavy metal exposure impairs oral fungal interactions in the population. This study extends our understanding of the biodiversity of oral fungi in the population and provides new insights for further studies on the factors influencing oral fungal homeostasis.

## 1. Introduction

Recently, human oral microbiota have emerged as an important research target of interest and study focus due to the role that the oral microbiome may play in the progression, diagnosis, and treatment of diseases [1]. The oral cavity is colonized by a large number of microorganisms, including viruses, protozoa, fungi, archaea, and bacteria. Within the human body, the complexity of the oral microbiota is second only to that of the colon [2]. Due to technical limitations, research on oral microorganisms in the past has mainly focused on bacteria, while fungi have remained understudied. In recent decades, methods for analyzing oral microorganisms have evolved rapidly with the shift from culturing to DNA detection and sequencing, which has significantly improved our understanding of oral fungi [3]. The composition of oral fungi is important for oral health. Native oral microbiota help to protect the host from invasion and subsequent infection by pathogenic microorganisms [4]. It has been shown that certain fungi may antagonize cariogenic bacteria in caries-free children [5]. It has also been shown that most oral fungi are associated with at least one bacterium or fungus that plays a stabilizing role in the organization of the overall microbial community [6]. Additionally, the disturbance of oral fungal homeostasis leads to an increase in pathogenic colonies, further increasing susceptibility to opportunistic infection [7]. Hei et al. confirmed the presence of disordered oral fungal communities in the oral cavity of gastric cancer patients and discovered the great potential of *M. globosa* in saliva and the tongue as a non-invasive biomarker for detecting gastric cancer [8]. Sajid et al. reported that smokeless tobacco intake is associated with the dysbiosis of oral fungal communities, and that such alterations may contribute to oral cancer [9]. Previous studies have also shown that oral fungi have been implicated in a variety of diseases, such as inflammatory bowel syndrome (IBS), CD, chronic respiratory diseases, and hepatitis B [10].

The structure of oral fungi is affected by several factors. For instance, alcohol consumption, marijuana usage, smoking, and outdoor environments [11]. Heavy metals are widely found in various environments such as soil, water, and plants. Smelting and mining operations are the most prominent anthropogenic sources of heavy metals [12,13]. Heavy metals in soil can enter the human body through the food chain, via oral intake or skin contact. Heavy metal pollution not only jeopardizes the human nervous system, blood system, and kidney function, but also affects the structure of the microbial community [14]. Davis et al. found that the salivary microbiome composition can change following exposure to antimony (Sb), arsenic (As), and mercury (Hg), which may be due to the toxicity of heavy metals to various human systems [15]. Zhang et al. revealed that long-term exposure to heavy metals from soil affects salivary microbial communities, which significantly alters the relative abundance of *Capnocytophaga*, *Selenomonas*, *Aggregatibacter*, and *Campylobacter* [16]. The study also showed an association between toxic metal exposure and oral microbiome-mediated diseases, for instance, dental caries is associated with increased levels of Sb and an increased abundance of *Lactobacillus* [17]. Although numerous studies have investigated the effects of environmental heavy metal exposure on oral microorganisms in recent years, most have been limited to bacteria. However, fungal communities are also an important component of the oral microbiota, and the investigation of oral fungi is significant for us to gain a deeper understanding of the effects of heavy metal exposure on the oral cavity.

FUNGuild is a tool that can be used to classify fungal OTUs based on ecological guild. There has already been research applying it in order to analyze the composition of oral fungi [9,18]. A group of species, whether related or unrelated, that utilize the same class of environmental resources in a similar way, are considered a guild [19]. Based on trophic strategies, fungi are divided into several functional groups. Combining these different trophic types of fungi forms the functional diversity of fungal communities, which is critical for maintaining health. Additionally, complex interactions among fungi can be better characterized by network analysis. This analysis is robust and capable of computing network topological properties, also revealing the keystone microbial group [20]. Keystone taxa are important species that have a significant impact on the microbial community structure, or even drive the covariation of the community. It has also been shown that keystone taxa play a particularly important role in the stability of their ecosystems [21]. Accordingly, identifying the keystone taxa of a fungal community contributes to a better understanding of the structure and functional diversity of the community.

In conclusion, the relationship between heavy metal exposure and oral fungal composition has not been comprehensively studied. The investigation of fungal communities can inform strategies to improve diagnostic tools and develop targeted therapies. To fill this gap, we studied oral fungi from different regional populations using Illumina high-throughput sequencing technology. Differences in oral fungal community structure and complex interactions were further analyzed using the FUNGuild software and ecological network analysis. Finally, we analyzed the effects of heavy metal exposure on the composition of human oral fungal communities to investigate the relationship between oral fungal community structure and human health. This study could contribute to promoting early disease intervention and reducing the burden of oral fungi-related diseases.

## 2. Materials and Methods

### 2.1. Study Area and Soil Sample Collection

The research sites are located in Baiyin (36°29′ N, 104°17′ E) and Lanzhou City (35°46′ N, 104°1′ E), Gansu Province, northwestern China. Due to long-term mining and metal smelting, the soil and air of Baiyin City have been contaminated with heavy metals [22,23,24]. For comparison, we selected Yuzhong County in Lanzhou City, which is geographically and climatically very close to Baiyin, as the control area. The local area is not contaminated with heavy metals.

A total of 13 sampling sites were set up in this study, including 6 in the exposure group and 7 in the control group. The sampling sites were situated in agricultural soils near local villages. Topsoil was collected in October 2021. According to the Technical Specification for Soil Environmental Monitoring (HJ/ST166-2004) [25], at each sampling site, a piece of farmland with an area of about 10 m × 10 m was randomly selected, and five sub-sampling sites were set up in each selected farmland using the five-point sampling method. Soil was collected at a depth of approximately 20 cm with a wooden shovel and well mixed with a sample weight greater than 1 kg. Soil samples were sealed in polyethylene bags and sent to the laboratory within 24 h. The soil was homogenized with a 2 mm mesh sieve to remove stones, roots, and other debris, and stored at 4 °C for soil property analysis.

### 2.2. Soil Heavy Metal Analysis

Soil samples of about 0.5 g were digested with 2 mL HCl, 6 mL HNO_3_, and 2 mL HF in a microwave digestion system (Milestone ETHOS ONE, Milestone, Milan, Italy). Then, inductively coupled plasma–mass spectrometry (ICP–MS, Agilent, Santa Clara, CA, USA) was used to determine the content of heavy metals (Mo, Cd, Sb, Cu, Zn, Hg, and Pb). Quality assurance/control procedures were performed using standardized reference materials (National Institute of Metrology, Beijing, China) for each batch of samples (one blank and one standard).

### 2.3. The Collection of Oral Buccal Mucosa Sample

In total, 92 and 44 participants were recruited from September 2019 to January 2021 from Baiyin and Lanzhou, respectively. The participants were all from villages near the agricultural soils and between the ages of 42–72, have lived in the local area permanently for at least 10 years, and have not been away from the local area for more than half a year continuously. All participants fulfilled the following four criteria: (i) subjects signed informed consent and had not used any antibiotics for at least 3 months prior to sampling; (ii) subjects had no oral diseases, such as halitosis, chronic xerostomia, untreated dental caries, abscesses, cancer, or candidiasis; (iii) subjects who reported being ill or unwell on the day of sampling were excluded; and (iv) subjects had at least 24 teeth. Prior to sample collection, participants were restricted from drinking or eating and were asked to gargle with water 30 min before sample collection. Then, we used a sterile cotton-wool swab to scrape the buccal mucosa on both the right and left sides of their mouths for 10 s. The project has been approved by ethical review.

### 2.4. Oral Microbial DNA Extraction and Sequencing Data Processing

Total microbial genomic DNA was extracted from each buccal mucosa sample using the E.Z.N.A.^®^ Tissue DNA Kit (Omega Bio-tek, Norcross, GA, USA) according to the manufacturer’s instructions. The quality and concentration of DNA were determined via 1.0% agarose gel electrophoresis and a NanoDrop^®^ ND-2000 spectrophotometer (Thermo Scientific Inc., Waltham, MA, USA), and the DNA samples were kept at −80 °C prior to further use. The ITS region of the fungal rRNA genes was amplified using primers ITS1F (5′-CTTGGTCATTTAGAGGAAGTAA-3′) and ITS2R (5′ -GCTGCGTTCTTCATCGATGC-3′) [26] via an ABI GeneAmp^®^ 9700 PCR thermocycler (ABI, Los Angeles, CA, USA). The PCR reaction mixture included 4 μL of 5 × Fast Pfu buffer, 2 μL of 2.5 mM dNTPs, 0.8 μL of each primer (5 μM), 0.4 μL of Fast Pfu polymerase, 10 ng of template DNA, and ddH_2_O to a final volume of 20 µL. The PCR amplification cycling conditions were as follows: initial denaturation at 95 °C for 3 min, followed by 27 cycles of denaturing at 95 °C for 30 s, annealing at 55 °C for 30 s, extension at 72 °C for 45 s, single extension at 72 °C for 10 min, and end at 4 °C. The PCR product was extracted from 2% agarose gel and purified using the AxyPrep DNA Gel Extraction Kit (Axygen Biosciences, Union City, CA, USA) according to the manufacturer’s instructions, and quantified using the Quantus™ Fluorometer (Promega, Madison, WI, USA).

Purified amplicons were pooled in equimolar amounts and paired-end sequenced on an Illumina MiSeq PE300 platform/NovaSeq PE250 platform (Illumina, San Diego, CA, USA) according to the standard protocols of Majorbio Bio-Pharm Technology Co. Ltd. (Shanghai, China). The resulting sequences were processed using the QIIME2 (version 2021.2) pipeline [27], quality-filtered using fastp version 0.19.6, and merged by FLASH version 1.2.11 based on the following criteria: (i) the 300 bp reads were truncated at any site receiving an average quality score of <20 over a 50 bp sliding window, the truncated reads shorter than 50 bp were discarded, and reads containing ambiguous characters were also discarded; (ii) only overlapping sequences longer than 10 bp were assembled according to their overlapped sequence. The maximum mismatch ratio of the overlap region is 0.2. Reads that could not be assembled were discarded. (iii) Samples were distinguished according to the barcode and primers, and the sequence direction was adjusted to exact barcode matching and 2-nucleotide mismatch in primer matching. Then, the optimized sequences were clustered into operational taxonomic units (OTUs) using UPARSE 7.1 with a 97% sequence similarity level [28]. The most abundant sequence for each OTU was selected as a representative sequence. OTUs with fewer than two sequences were eliminated, and their representative sequences were assigned to taxonomic lineages using the RDP classifier version 2.2 against the SILVA database (version 138) using a confidence threshold of 0.7.

### 2.5. Statistical Analysis

The alpha diversity indices were calculated using mothur software (http://www.mothur.org/wiki/Calculators, accessed on 27 January 2025, version 1.45.3). Principal coordinate analysis (PCoA) was used to evaluate the influence of heavy metal exposure on the structure of microbial communities. FUNGuild software (http://www.funguild.org/, accessed on 27 January 2025, version 1.0) was used for an analysis of the guild of each OTU. Data with absolute values of Spearman correlation coefficient (r) > 0.6 and Benjamini and Hochberg false discovery rate (FDR)-corrected *p*-values < 0.05 were selected to construct the co-occurrence network. Network images were generated using Cytoscape 3.9.1 and the network properties (i.e., average degree, average clustering coefficient, and density) were calculated, which characterize the degree of densification and clustering of the network. A Wilcoxon Mann–Whitney test was performed using the IBM SPSS 26.0 software to compare the α-diversity indices between groups, and *p* < 0.05 was defined as significant. Chi-square tests and *t*-tests were performed using the IBM SPSS 26.0 software to characterize the participants.

## 3. Results

### 3.1. Fungal Community Composition

The oral samples were collected from Baiyin City and Lanzhou City, Gansu Province. There were significant differences in the heavy metal concentrations of the agricultural soils between the two regions (Table 1). There was no significant difference in gender, age, smoking, or alcohol consumption between the two study groups (Appendix A). Illumina high-throughput sequencing of fungal ITS1 rDNA from 136 samples resulted in 8,939,868 high-quality fungal sequences. OTU clustering based on 97% sequence similarity cutoff yielded 14 phyla, 48 classes, 122 orders, and 685 genera. These data indicated that fungal communities are abundant in the oral cavity. The 136 samples were divided into two groups based on heavy metal concentrations, with the Baiyin area as the exposure group and the Lanzhou area as the control group. The common and unique species of the two groups were compared using Venn diagrams. The results are shown in Appendix A. There were 516 identical OTUs in the two groups. The exposure group had 1254 unique OTUs and the control group had 579 unique OTUs. There were more OTU species unique to the exposure group than to the control group, which may be due to heavy metal exposure affecting the stabilization of oral fungi.

We selected the top ten phyla of relative abundance to analyze the composition of the fungal communities in the two groups (Figure 1). Among them, the top three fungi of the exposure group in order were Basidiomycota, Ascomycota, and Rozellomycota. The relative abundances were 68.21%, 27.85%, and 0.34%, respectively. In contrast, the top three fungi in the control group were Ascomycota, Basidiomycota, and Mortierellomycota, in descending order. The relative abundances were 56.84%, 40.13%, and 0.31%, respectively. The Ascomycota, Basidiomycota, and Mortierellomycota compositions were significantly different in the two groups. In addition, Chytridiomycota was significantly different in the two groups, with relative abundances of 0.06% and 0.16% in the exposure and control groups, respectively. The relative abundance of other fungal phyla was lower than 0.10% in both groups, with no significant difference.

The relative abundance of oral fungi at the genus level is shown in Figure 2. Only the top 15 genera of relative abundance were shown in the figure. *Apiotrichum* was the genus with the highest relative abundance in the exposure group (42.79%) and the second most abundant genus in the control group (21.28%). *Cutaneotrichosporon* ranked second in the exposure group and third in the control group (21.39% and 11.94, respectively). The genus with the highest abundance was *Aspergillus*, with a relative abundance of 35.05% in the control group, but its relative abundance in the exposure group was only 3.74%. The relative abundance of the above three genera was significantly different in the two groups. The relative abundance of the other genera ranged from 0.09% to 2.89%, which was significantly different for *Debaryomyces*, *Wallemia*, and *Saccharomyces* in the two groups. It has been reported that *Cutaneotrichosporon* is often associated with human hosts, and may be an opportunistic human pathogen [29]. Therefore, in order to more accurately identify the causative fungi, we analyzed the genera *Apiotrichum* and *Cutaneotrichosporon* at the species level (Appendix A). The relative abundances of *Apiotrichum montevideense*, *Cutaneotrichosporon curvatus*, and *Cutaneotrichosporon cutaneum* were significantly higher in the exposure group than in the control group.

### 3.2. Diversity Analysis of Fungal Communities

Three α-diversity indices (Sobs, Ace, and Chao1) were used to evaluate the richness and diversity of fungal communities (Figure 3A). The results showed that the fungal community richness was significantly higher in the control group than in the exposure group.

Principal coordinate analysis (PCoA) based on the Bray–Curtis (Figure 3B) distance showed that the PC1 axis accounted for about 34.97% of the total variation, the PC2 axis accounted for about 13.69% of the total variation, and the two axes accounted for a total of 48.66% of the variation, which explained nearly half of the variation. This suggested a significant difference in the distribution of fungal communities between the two groups (ANOSIM = 0.3900, *p* = 0.001).

Based on the above results, heavy metal contamination of the soil decreased the oral fungal abundance of local populations, with a significant effect on fungal community composition.

### 3.3. The Trophic Modes of Fungal Communities

In order to analyze the trophic strategies of fungal communities, the annotation tool FunGuild was used to classify fungal OTUs into specific trophic taxa, then further subdivide them into specific ecological guilds.

FUNGuild detected eight major trophic modes and eighteen guilds between the two groups (Figure 4A). A total of 916 OTUs were predicted as highly probable and probable life strategies out of 2351 OTUs, which amounts to 38.96% of the total OTUs. The top three trophic modes in the exposure group were 48.48% saprotroph, 21.82% pathotroph, and 21.04% pathotroph–saprotroph, respectively. In total, this accounted for 91.34% of all trophic modes. The top three trophic modes in the control group were the same as the exposure group, 56.14% saprotroph, 20.05% pathotroph, and 14.92% pathotroph–saprotroph, which amounted to 91.11%. The five other trophic modes accounted for less than 10% of both groups, with symbiotroph (5.14% and 4.74%) being the highest proportion of the five. Saprotroph, pathotroph, and pathotroph–saprotroph were the major trophic modes in oral fungi. The predominant trophic modes were differently represented in the two groups, which might be due to heavy metal exposure. The above results indicated that the composition of trophic modes was similar in both groups of fungal communities. Interestingly, compared to the control group, the proportion of saprotroph decreased and the proportion of pathotroph–saprotroph increased in the exposure group. This suggests that these fungi had different susceptibilities to heavy metals. The proportion of shared and exclusive species within the top three trophic modes was similar in the exposure group (Figure 5). The exclusive OTUs in the exposure group were greater than the shared OTUs, which suggests that heavy metal exposure may have disrupted the homeostasis of oral fungi, leading to the entry of more exogenous fungi [4]. We concluded that the species compositions of the two groups were different.

The composition of the guild in both groups was analyzed. Undefined saprotroph was the largest guild in sequence richness in the two groups (Figure 4B). The proportions were 40.38% and 43.48% in the exposure and control groups, respectively. Plant Pathogen was the second largest guild, with 11.44% and 9.41%, respectively, and belongs to the pathotroph. At over 50% together, these two guilds were the major fungal guilds. The most numerous guild within pathotroph-saprotroph was Plant Pathogen–Wood Saprotroph, which accounted for just 3.07% and 1.76% of the two groups.

At the genus level, the trophic modes of species that differed in relative abundance were saprotroph, pathotroph, and pathotroph–saprotroph–symbiotroph (Appendix A). Five fungi belonged to saprotroph, four of which had the guild of Undefined saprotroph. The relative abundance in the exposure group was significantly lower than that in the control group. This is consistent with the above trend of Undefined saprotrophs in the two groups. The guild of *Apiotrichum* belongs to Soil saprotroph and *Cutaneotrichosporon* belongs to Animal Pathogen in pathotroph, which had a higher relative abundance in the exposure group than in the control group. *Vishniacozyma* showed the opposite trend, it belongs to pathotroph–saprotroph–symbiotroph, a trophic mode with a very low proportion of the entire sequence. The guild was Fungal Parasite–Undefined saprotroph.

### 3.4. Patterns of Fungal Ecological Networks

To further analyze the relationship between the fungi in the different groups, fungal ecological network analysis was performed based on OTU data. The topological characteristics of the two groups of fungal ecological networks are shown in Table 2. The network degrees of the paired fungal nodes in both groups were power-law distributed, which indicated that these co-occurring networks were reliable and non-randomized [30]. The network size of the control group was significantly larger than that of the exposure group. The number of nodes for the control group was 135, whilst for the exposure group, it was 26. The number of links in the control group was 224, while in the exposure group, it was 24. The average degree showed that there was a difference in the complexity of the network between the two groups; average degree was 1.846 for the exposure group and 3.318 for the control group. The average clustering coefficients of the two groups were 0.127 and 0.202, respectively. In conclusion, the fungal ecological network of the control group was more complex and stable.

The nodes were categorized into four classes based on within-module connectivity (*Zi*) and between-module connectivity (*Pi*): peripherals (*Zi* ≤ 2.5, *Pi* ≤ 0.62), connectors (*Zi* ≤ 2.5, *Pi* ≥ 0.62), module hubs (*Zi* ≤ 2.5, *Pi* ≥ 0.62), and network hubs (*Zi* ≥ 2.5, *Pi* ≥ 0.62). Among them, connectors, module hubs, and network hubs were defined as keystone taxa (Figure 6A). Keystone taxa were not detected in the exposure group. One connector was detected in the control group, *Filobasidium_globisporum*, which belongs to saprotroph. Seven module hubs were detected, of which *Kodamaea_ohmeri*, *Humicola_nigrescens*, and *Kazachstania_servazzii* belong to saprotroph, *Dothidea_insculpta* is pathotroph–saprotroph–symbiotroph, *Trichothecium_roseum* and *Trichosporon_coremiiforme* belong to pathotroph, and *Symmetrospora_coprosmae* was unassigned.

The statistics of these two network nodes are shown in Figure 6B. The fungal functional annotation results (Figure 6C) showed that the trophic modes of the two ecological networks were significantly different. The highest number of trophic mode nodes in both groups was for saprotroph, the second highest in the exposure group was for pathotroph, and the second highest in the control group was for pathotroph–saprotroph–symbiotroph. The number of nodes in the control group was much higher than that in the exposure group; there were 12 nodes shared between the two fungal networks (Figure 6D). In addition, robustness was chosen to measure the stability of the fungal ecological network. The results of the calculated robustness using both weighted and unweighted methods were consistent. The robustness of the fungal network differed significantly between the two groups and was higher in the control group than in the exposure group, which indicates that the fungal network in the control group was more stable (Figure 7).

## 4. Discussion

### 4.1. Effects of Heavy Metal Exposure on the Composition of Human Oral Fungal Communities

A complex microbial community exists in the oral cavity, including bacteria, fungi, and viruses. It has been shown that the diversity of oral microorganisms and oral health have a close relationship, and that the living environment of the host alters the susceptibility of the host to disease by influencing the composition, structure, and metabolic function of oral microorganisms [31]. In the present study, we analyzed the effects of heavy metal exposure on oral fungal communities by comparing the community composition, functional diversity, and network composition of oral fungi in populations from different areas. It was suggested that more fungal species were found in healthy populations compared to diseased populations, and that oral infections might be due to a reduction in microbiological diversity [32]. In this research, fungal abundance in the exposure group was lower than that in the control group, suggested that the exposure of soil to heavy metal may increase the risk of oral disease in the local population.

Also, there was a significant difference in the distribution of fungal communities between the two groups. At the phylum level, the relative abundance of Basidiomycota was 40.13% in the control group and increased to 68.21% in the exposure group. Ascomycota showed the opposite trend, with a relative abundance of 56.84% in the control group and 27.85% in the exposure group. At the genus level, the relative abundances of *Aspergillus*, *Debaryomyces*, *Wallemia*, *Saccharomyces*, and *Vishniacozyma* were much higher in the control group than in the exposure group. Previous studies have confirmed the beneficial effects of *Saccharomyces* in the prevention and treatment of diseases [33]. For example, treatment with *Saccharomyces Boulardii* contributed to the improvement of acute diarrhea syndrome caused by a virus [34]. It has also been shown that *Saccharomyces Boulardii* produces numerous bioactive metabolites that possess antioxidant, antibacterial, antitumor, and anti-inflammatory properties [35]. In the present study, heavy metal exposure reduced the relative abundance of *Saccharomyces*, which may lead to reduced immunity in local populations. *Apiotrichum* and *Cutaneotrichosporon* had a relative abundance of 42.79% and 21.39% in the exposure group, respectively, which was significantly higher than the 21.28% and 11.94% in the control group. *Apiotrichum* is widespread in nature, commonly found in soil, biogas reactors, and farm animals [36], as the main cause of clinical disease in the form of pulmonary infection [37,38]. *Cutaneotrichosporon* is a recently established genus [39], both it and *Apiotrichum* were previously classified within the genus *Trichosporon*. In the present study, the relative abundances of *Apiotrichum montevideense*, *Cutaneotrichosporon curvatus*, and *Cutaneotrichosporon cutaneum* were significantly higher in the exposure group than in the control group. It has been reported that *Apiotrichum montevideense* and *Cutaneotrichosporon cutaneum* were identified from patients with superficial infections [40]. Accordingly, we suggested that exposure to heavy metals may lead to an increase in the abundance of some pathogenic fungi. The above results suggest that heavy metal exposure may increase the risk of diseases associated with *Apiotrichum* and *Cutaneotrichosporon* infection in the local populations. In addition, the relative abundance of *Candida* was higher in the exposure group than in the control group, but there was no significant difference. Most of the current research on oral fungi is on *Candida*, which is the main cause of oral fungal infections [41]. Although the difference in *Candida* was not significant in this study, the increase in this fungus is also notable.

In conclusion, the dominant flora in the two groups was significantly different. Heavy metal exposure affected the composition of oral microorganisms in local populations. Decreased beneficial fungi and increased pathogenic fungi may increase the risk of disease in local populations.

### 4.2. Trophic Strategies and Co-Occurrence Network Analysis of Human Oral Fungal Communities

To further investigate the effects of heavy metal exposure on human oral fungi, the functional diversity and network composition of oral fungi in the two groups were analyzed. The FUNGuild results predicted that oral fungi majorly belong to saprotroph, and that the guild with the highest relative abundance in saprotroph was Undefined saprotroph. The different species analysis at the genus level showed that the fungal genera belonging to Undefined saprotroph were all lower in relative abundance in the exposure group than in the control group. However, it has been reported that Undefined saprotroph and heavy metals in soil were positively associated, which is inconsistent with the results of the present study on oral cavities [42]. We hypothesized that this is due to the fact that Undefined saprotroph includes multiple fungal species with different proportions in the soil and the oral cavity, which have different tolerances to heavy metals. The exact environmental requirements and ecological role of Undefined saprotroph are unknown and require further study. In contrast, the relative abundance of Soil saprotroph was lower in the exposure group than in the control group. The abundance of Soil saprotroph was related to soil aggregates’ stability, soil’s physical properties, and soil nutrients [43]. The relative abundance of the species in the oral cavity may be influenced by the heavy metal contamination and physicochemical properties of the local soil. *Cutaneotrichosporon* within pathotroph, an Animal Pathogen, had a higher relative abundance in the exposure group than in the control group. It has been reported that Animal Pathogens can cause disease in humans when animals are in close contact with humans, and their impact on socioeconomics, food safety and security, and human health has received widespread attention [44]. This is consistent with the former results in this study, which indicate that *Cutaneotrichosporon* may have detrimental effects on population health. The heavy metal contamination of soil may lead to an increase in the relative abundance of Animal Pathogens in the oral fungi of the local population, which may make the local population more susceptible to disease. In addition, *Vishniacozyma* of pathotroph–saprotroph–symbiotroph, which belongs to Fungal Parasite–Undefined saprotroph, had a lower abundance in the exposure group than in the control group, and the same abundance variation as Undefined saprotroph.

Ecological network analysis can be used to understand potential ecological associations between microbial communities and illustrate the effects of environmental heavy metal exposure on oral fungi in local populations. Compared to the exposure group, the complexity of the control group was mainly linked to total nodes, total links, and average degree. Eight keystone taxa were identified in the control group, while no keystone taxa were identified in the exposure group. The keystone taxa in the control group belong to Ascomycota and Basidiomycota at the phylum level, which are dominant phyla. The above results suggest that dominant phyla have greater potential to become keystone taxa, and that heavy metal exposure led to the loss of keystone taxa and imperiled the stability of the community. Simultaneously, robustness indicated that the control group was more stable than the exposure group, suggesting that the complex fungal network could increase the stability of the community and improve resistance to disturbance (Bello et al., 2021 [20]). In addition, we used trophic modes to further explain the co-occurrence network patterns of fungal communities. Most of the nodes observed in this study belonged to saprotroph, a group of microorganisms capable of obtaining carbon through the decomposition of complex organic matter, which is a key microorganism in regulating nutrient cycling in terrestrial ecosystems [45]. Compared to the exposure group, a higher percentage of saprotrophs in the control group participated in the co-occurrence network, which suggests that heavy metal exposure inhibited the growth of saprotrophs in this study. Meanwhile, the network of the control group had more pathotroph–saprotroph–symbiotroph. This may be due to the fact that heavy metal exposure reduced the interactions between oral fungi, resulting in a reduced proportion of symbiotic types in the network.

Each taxonomic unit of the fungus plays a different role in the community, and mapping to ecological networks implies that nodes have different topological functions [46]. According to the *Zi*-*Pi* plot, most of the fungi in the two groups were peripheral nodes. These fungi had few associations with other fungi. In the network of the control group, seven nodes belonged to connectors and one node belonged to module hubs. Connectors and module hubs were generalists in the fungal network and considered to be the keystone taxon with the greatest influence on the structure and potential function of the microbial community [47]. No keystone taxa appeared in the exposure group. Therefore, we suggest that heavy metal exposure impairs the community structure of oral fungi in local populations and that the impaired fungal network may lead to an increased incidence of disease [48]. The module hubs in the control group belonged to saprotroph, one node in the connectors belonged to saprotroph, and two belonged to pathotroph. This indicated that saprophytic and pathotrophic nutrition were the keystone functions of oral fungi in the control group. Additionally, keystone taxa have been reported to boost host immunity. Therefore, we suggested that the absence of keystone taxa in the network of the exposure group was due to the fact that heavy metal exposure weakened the immunity of the local population.

## 5. Conclusions

In the present study, we focused on the effects of heavy metal exposure in soil on the oral fungal communities of local populations. By using high-throughput sequencing data and microbial bioinformatics analysis methods, the species diversity, functional diversity, and corresponding fungal ecological networks of fungi in the oral cavity were investigated. The results show that heavy metal exposure significantly decreased the abundance of oral fungi in the local population and impaired the stability and complexity of the fungal network. Different oral fungi responded differently to heavy metal exposure. Heavy metal exposure increased the relative abundance of *Apiotrichum* and *Cutaneotrichosporon*, which might lead to an increased risk of disease in local populations. Further studies are required to determine the specific functions of the different species in the two groups. In addition, the nutritional strategies of fungi were similar in both groups, with saprophytic fungi being the dominant species in the oral cavity. *Cutaneotrichosporon* in pathotroph was an Animal Pathogen, with a higher relative abundance in the exposure group than in the control group, and it might result in an increased risk to human health. Saprophytic and pathotrophic fungi were the key microorganisms influencing the structure and potential function of the fungal community in the control group. The network size in the exposure group was smaller than that in the control group, and no keystone taxa were observed. This might be due to a reduction in the interactions between oral fungi as a result of heavy metal exposure. A decreased number of oral fungal interactions can lead to more fragile communities and an increased risk of disease. These results may expand our understanding of human oral fungal biodiversity and the impact of heavy metal exposure on the survival strategies of oral fungi in local populations, which has not yet been extensively studied.

## Figures and Tables

**Figure 1 microorganisms-13-00622-f001:**
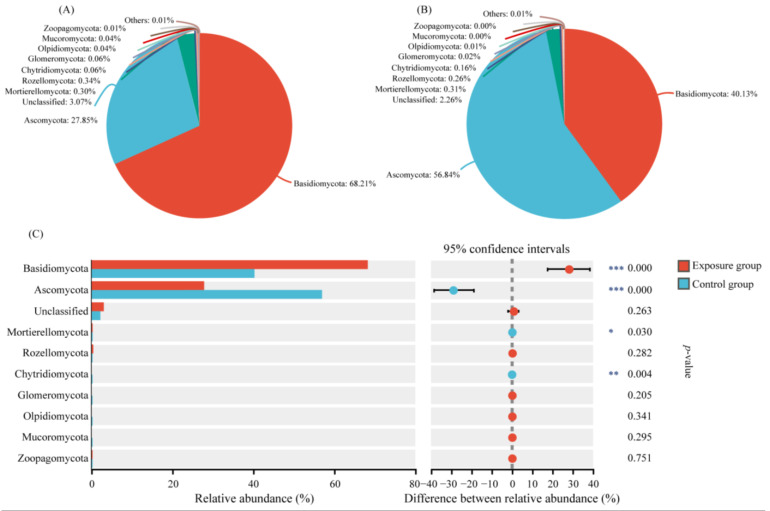
Variation in fungal communities at the phylum level in the two groups. (**A**) Exposure group; (**B**) control group; and (**C**) species analysis of phylum level differences between the two groups (*** *p* < 0.001, ** 0.001 < *p* < 0.01, and * 0.01 < *p* < 0.05).

**Figure 2 microorganisms-13-00622-f002:**
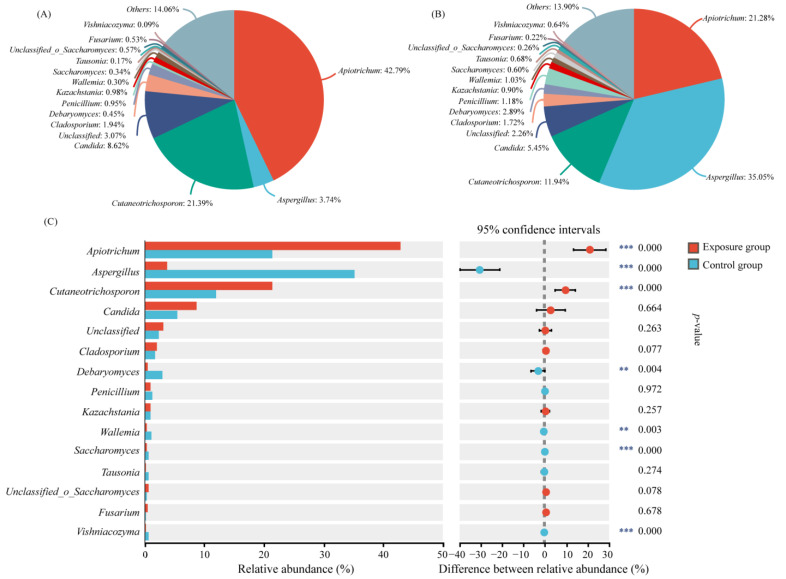
Variation in fungal communities at the genus level in the two groups. (**A**) Exposure group; (**B**) control group; and (**C**) species analysis of genus level differences between the two groups (*** *p* < 0.001 and ** 0.001 < *p* < 0.01).

**Figure 3 microorganisms-13-00622-f003:**
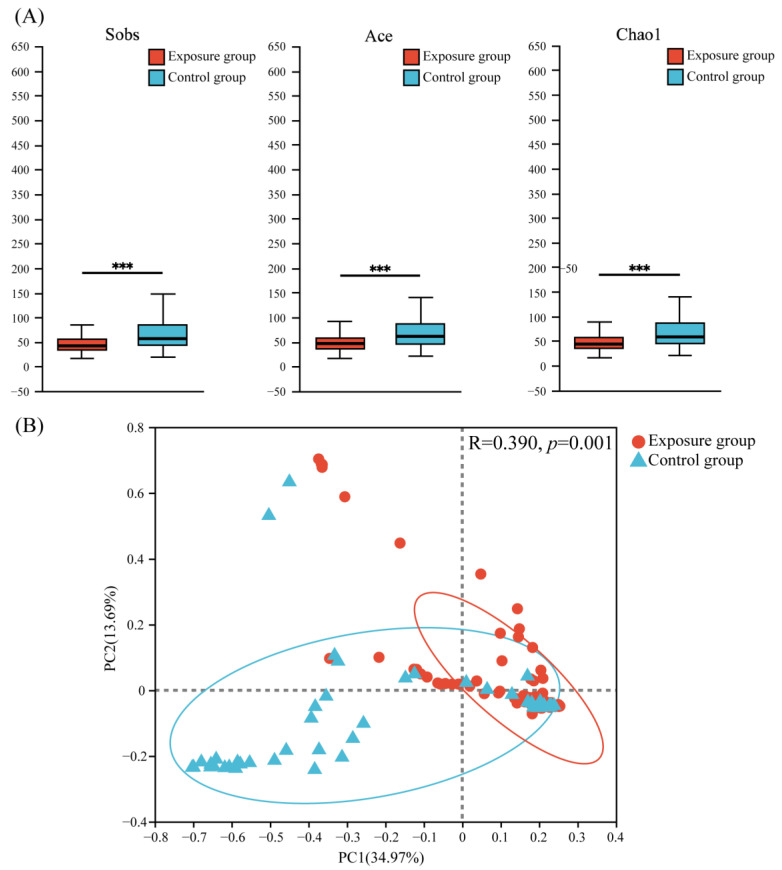
(**A**) Alpha diversity and (**B**) principal coordinate analysis (PCoA) of fungal communities in the two groups (*** *p* < 0.001).

**Figure 4 microorganisms-13-00622-f004:**
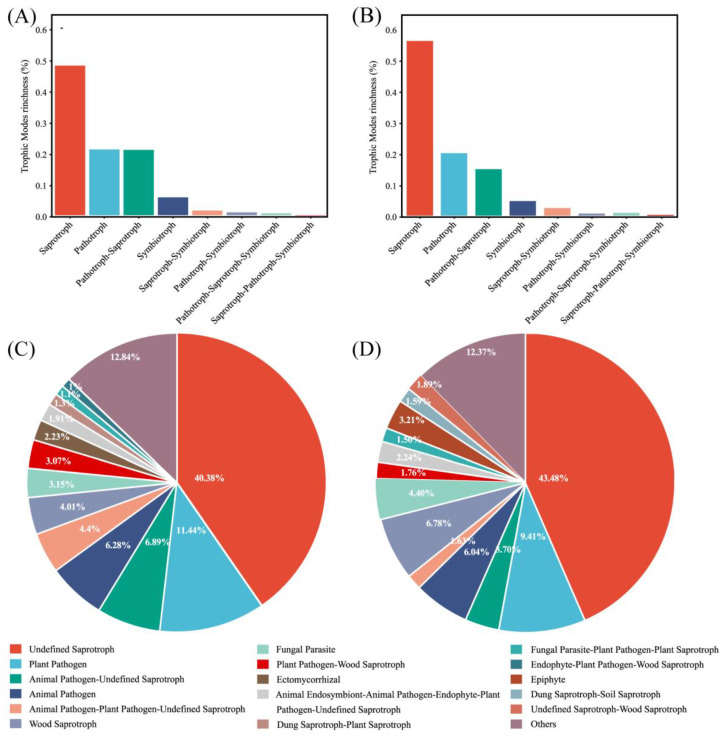
Guild assignment of two groups with OTU data using FUNGuild software. (**A**) Trophic modes for the exposure group; (**B**) trophic modes for the control group; (**C**) guild for the exposure group; and (**D**) guild for the control group.

**Figure 5 microorganisms-13-00622-f005:**
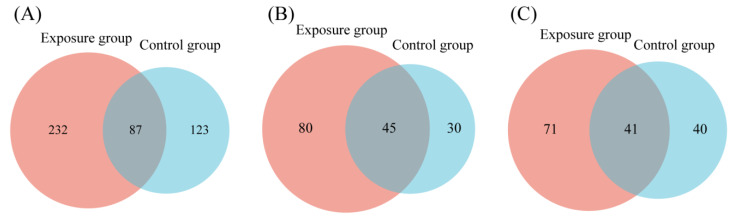
Venn diagrams comparing the number of oral fungal OTUs in two groups. (**A**) Saprotroph; (**B**) pathotroph; and (**C**) pathotroph–saprotroph–symbiotroph.

**Figure 6 microorganisms-13-00622-f006:**
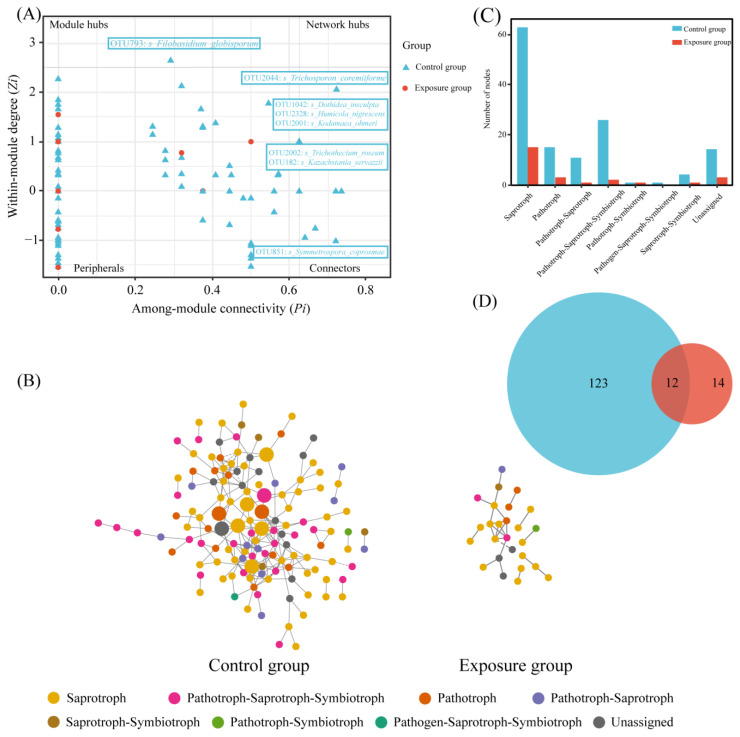
Network analysis of fungal communities in two groups. (**A**) *Zi*-*Pi* plot; (**B**) co-occurrence network of fungal communities; (**C**) the number of trophic modes in the two fungal networks; and (**D**) Venn diagram of fungi at the genus level in the two fungal networks.

**Figure 7 microorganisms-13-00622-f007:**
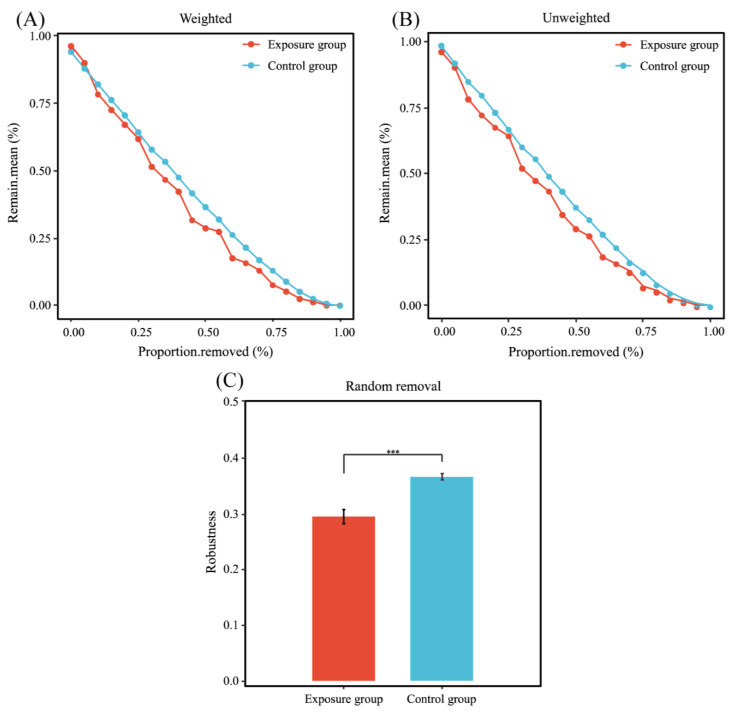
Stability analysis of co-occurrence networks. (**A**) Weighted; (**B**) unweighted; and (**C**) robustness (*** *p* < 0.001).

**Table 1 microorganisms-13-00622-t001:** Heavy metal content at sampling locations.

	Exposure Group	Control Group	*p*-Value
Mo (ng/g)	0.22 ± 0.16	0.08 ± 0.03	0.001
Cd (ng/g)	8.42 ± 14.09	0.10 ± 0.04	0.001
Sb (ng/g)	0.02 ± 0.01	0.00 ± 0.00	0.001
Cu (ng/g)	117.33 ± 53.13	17.38 ± 3.89	0.001
Zn (ng/g)	420.23 ± 709.09	47.47 ± 10.16	0.001
Hg (ng/g)	0.42 ± 0.38	0.02 ± 0.02	0.002
Pb (ng/g)	150.88 ± 146.13	12.15 ± 2.57	0.001

*p*-value: the significance between exposure and control groups compared with the two-tailed Wilcoxon rank-sum test.

**Table 2 microorganisms-13-00622-t002:** Topological characteristics of oral fungal co-occurrence networks.

Network Indices	Exposure	Control	Exposure	Control
	Empirical		Random	
Total nodes	26	135	n.a	n.a
Total links	24	224	n.a	n.a
RMT cut-off	0.86	0.86	n.a	n.a
R square of power-law (R^2^)	0.905	0.817	n.a	n.a
Average clustering coefficient (avgCC)	0.127	0.202	0.007 +/− 0.019	0.015 +/− 0.011
Average path distance (GD)	2.558	4.634	3.017 +/− 0.358	3.927 +/− 0.094
Geodesic efficiency (E)	0.536	0.269	0.434 +/− 0.042	0.296 +/− 0.005
Harmonic geodesic distance (HD)	1.867	3.719	2.326 +/− 0.226	3.376 +/− 0.06
Centralization of degree (CD)	0.126	0.072	0.126 +/− 0	0.072 +/− 0
Centralization of betweenness (CB)	0.086	0.19	0.234 +/− 0.094	0.14 +/− 0.024
Centralization of stress centrality (CS)	0.162	0.426	0.017 +/− 0.007	0.056 +/− 0.031
Centralization of eigenvector centrality (CE)	0.828	0.894	0.778 +/− 0.032	0.823 +/− 0.019
Centralization of closeness centrality (CCL)	0.033	0.015	0.056 +/− 0.018	0.046 +/− 0.038
Density (D)	0.074	0.025	0.074 +/− 0	0.025 +/− 0
Reciprocity	1	1	1 +/− 0	1 +/− 0
Transitivity (Trans)	0.409	0.278	0.082 +/− 0.057	0.045 +/− 0.012
Connectedness (Con)	0.265	0.726	0.52 +/− 0.117	0.911 +/− 0.043
Efficiency	0.81	0.975	0.992 +/− 0.005	0.994 +/− 0
Hierarchy	0	0	0.074 +/− 0	0.025 +/− 0
Lubness	1	1	0.446 +/− 0.15	0.226 +/− 0.043
Modularity	0.629	0.662	0.602 +/− 0.028	0.537 +/− 0.01

n.a: No data available.

## Data Availability

The datasets presented in this study can be found in online repositories. The raw sequence data have been deposited in the NCBI Sequence Read Archive under BioProject accession number PRJNA1084108.

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
