# Peer review of "Effects of Long-Term Heavy Metal Exposure on the Species Diversity, Functional Diversity, and Network Structure of Oral Mycobiome"

_microorganisms, 2025, doi:10.3390/microorganisms13030622_

Round 1
Reviewer 1 Report
Comments and Suggestions for Authors
Dear Authors,
Congratulations on your well-designed and interesting study. The topic of oral fungal homeostasis and how it relates to heavy metal exposure is very relevant and timely, given the increasing concerns about environmental pollution and its impact on human health. Your study presents intriguing new data on the effect of chronic heavy metal exposure on oral fungi, with significant differences in species richness, abundance, and fungal network stability between exposed and control populations.
Sincerely
Author Response
Comments 1: Congratulations on your well-designed and interesting study. The topic of oral fungal homeostasis and how it relates to heavy metal exposure is very relevant and timely, given the increasing concerns about environmental pollution and its impact on human health. Your study presents intriguing new data on the effect of chronic heavy metal exposure on oral fungi, with significant differences in species richness, abundance, and fungal network stability between exposed and control populations.
Response 1:Thanks very much for the time and effort that you have put into reviewing the previous version of the manuscript. We thank you for your positive comment about the originality and quality of this work and your appreciation of the importance of the findings reported in our manuscript.
Reviewer 2 Report
Comments and Suggestions for Authors
Dear Authors,
I have read the article and believe that some additions are needed to improve the clarity, structure, and accuracy of the information presented:
Abstract: neither the method used, nor the conclusions are specified.
Introduction: More details are needed regarding the importance of the oral microbiome not only in determining disease but also in preventing it.
The link between heavy metal exposure and damage to the structure of the oral microbiome should be more clearly stated.
The purpose of the study is not clear in the last paragraph. It would be necessary to explain the importance of this study in the current context of oral microbiome research and public health.
Material and method: How was the depth from which the samples were collected determined? How many samples were collected?
Why was the E.Z.N.A.® Soil DNA Kit specifically chosen for DNA extraction from oral mucosa and not another kit? What are the advantages of using it?
What criteria were used to evaluate the quality of the obtained DNA?
It should be explained why the process of clustering OTUs at the 97% similarity level is important.
It should be explained how the network properties (average degree, average clustering coefficient, and density) are interpreted.
Results and discussions: Should it be explained more clearly why these differences in the relative abundance of the genera are significant, i.e., a higher or lower abundance of genera in one group compared to another?
What is the RANOSIM test, and how does it explain the distribution of fungal communities between groups?
How might heavy metal exposure influence observed changes in the proportion of saprotrophs and pathotrophs between groups?
What is the significance of the fact that there are several exclusive OTUs in the exposure group?
What does each category of nodes represent in the analysis of ecological networks (for example, "connector" type nodes or "network hubs"? What impact do these categories have on the network structure and its stability?
What is the significance of the fact that no key taxa were detected in the exposure group? Does this influence the interpretation of fungal network stability?
It should be more clearly explained what the saprotrophic mode means, what role it plays in the oral ecosystem, and why exposure to heavy metals appears to inhibit the growth of Saprotrophs.
How the changes in the networks influence the ecological balance of the oral microbiome should be detailed.
What are the limitations of this study?
Author Response
Thank you very much for taking the time to review this manuscript. We have significantly improved the manuscript according to the suggestions, which we believe has very much improved our paper. And detailed corrections are listed point by point. Also, we have highlighted the corresponding changes in the article.
Comments 1: Abstract: neither the method used, nor the conclusions are specified.
Response 1: We sincerely appreciate the valuable comments. We have modified the abstract so that it adequately presents the results of our study. And it now reads (Page1, lines 11-31):
Oral fungal homeostasis is closely related to the state of human health, and its composition is influenced by various factors. At present, the effects of long-term soil heavy metal exposure on oral fungi of local populations have not been adequately studied. In this study, we used inductively coupled plasma–mass spectrometry (ICP-MS) to detect heavy metals in agricultural soils from two areas in Gansu Province, northwestern China. ITS amplicon sequencing was used to analyze the community composition of oral buccal mucosa fungi from local village residents. Simultaneously, functional annotation of fungi was performed using FUNGuild, and co-occurrence networks were constructed to analyze the interactions of different functional fungi. The results showed that the species diversity of oral fungi of local populations in the soil heavy metal exposure group was lower than that of the control population. The relative abundance of Apiotrichum and Cutaneotrichosporon was higher in the exposure group than in the control group. In addition, Cutaneotrichosporon is an Animal Pathogen, which may lead to an increased probability of disease in the exposure group. Meanwhile, there were significant differences in co-occurrence network structure between the two groups. The control group had a larger and more stable network than the exposure group. Eight keystone taxa were observed in the network of the control group, while none were observed in the exposure group. In conclusion, heavy metal exposure may increase the risk of diseases associated with Apiotrichum and infection Cutaneotrichosporon in the local populations. It can also lead to loss of keystone taxa and reduced stability of the oral fungal network. The above results illustrated that heavy metal exposure impairs the oral fungal interactions in the population. This study extends our understanding of the biodiversity of oral fungi in the population and provides new insights for further studies on the factors influencing oral fungal homeostasis.
Comments 2: Introduction: More details are needed regarding the importance of the oral microbiome not only in determining disease but also in preventing it.
Response 2: We would like to thank you for your constructive comments. We have added this information to the article (Page 2, lines 45-51):
The composition of oral fungi is important for oral health. Native oral microbiota helps protect the host from invasion and subsequent infection by pathogenic microorganisms (Sato, et al., 2021). It has been shown that certain fungi may antagonize cariogenic bacteria in caries-free children (Diaz and Dongari-Bagtzoglou, 2021). It has also been shown that most oral fungi are associated with at least one bacterium or fungus which plays a stabilizing role in the organization of the overall microbial community (Cheung, et al., 2022).
Comments 3: The link between heavy metal exposure and damage to the structure of the oral microbiome should be more clearly stated.
Response 3: Thank you for your feedback. We have added this information to the article (Page 2, lines 69-77):
Davis et al. found that salivary microbiome composition can change following exposure to antimony (Sb), arsenic (As) and mercury (Hg), which may be due to the toxicity of heavy metals to various human systems. (Davis, et al., 2020). Zhang et al. revealed that long-term exposure to soil heavy metals affects salivary microbial communities, which significantly altered the relative abundance of Capnocytophaga, Selenomonas, Aggregatibacter, and Campylobacter. (Zhang, et al., 2022). The study also showed an association between toxic metal exposure and oral microbiome mediated diseases, for instance, dental caries is associated with increased levels of Sb and increased abundance of Lactobacillus. (Adler, et al., 2021).
Comments 4: The purpose of the study is not clear in the last paragraph. It would be necessary to explain the importance of this study in the current context of oral microbiome research and public health.
Response 4: We sincerely appreciate the valuable comments. We have added this information to the article (Page 2, lines 98-100, 104-107):
In conclusion, the relationship between heavy metal exposure and oral fungal composition has not been well-studied. Investigation of fungal communities can inform strategies to improve diagnostic tools and develop targeted therapies.
Finally, we analyzed the effects of heavy metal exposure on the composition of human oral fungal communities to investigate the relationship between oral fungal community structure and human health. This study could contribute to promoting early disease intervention and reducing the burden of oral fungal-related diseases.
Comments 5: Material and method: How was the depth from which the samples were collected determined? How many samples were collected?
Response 5: Thank you for your comment. In this study, we chose topsoil for analysis, thus the depth of collection was 20 cm, and a total of 13 soil samples were collected, 6 from the exposure group and 7 from the control group. We have added this information to the article (Page 3, lines 116-123):
A total of 13 sampling sites were set up in this study, including 6 in the exposure group and 7 in the control group. Sampling sites were situated in agricultural soils near local villages. Topsoil was collected in October 2021. According to the Technical Specification for Soil Environmental Monitoring (HJ/ST166-2004), at each sampling site, a piece of farmland with an area of about 10 m × 10 m was randomly selected, and five sub-sampling sites were set up in each selected farmland using the five-point sampling method. Soil was collected at a depth of approximately 20 cm with a wooden shovel and well mixed with a sample weight greater than 1 kg.
Comments 6: Why was the E.Z.N.A.® Soil DNA Kit specifically chosen for DNA extraction from oral mucosa and not another kit? What are the advantages of using it?
Response 6: Thank you for bringing this to our attention. We are very sorry for our writing error, “soil” should be “Tissue”, the error has been corrected in the article (Page 4, line 149). E.Z.N.A.® Tissue DNA Kit is used to extract DNA from a wide range of tissues including oral cavity.
Total microbial genomic DNA was extracted from each buccal mucosa samples using the E.Z.N.A.® Tissue DNA Kit (Omega Bio-tek, Norcross, GA, U.S.) according to manufacturer’s instructions.
Comments 7: What criteria were used to evaluate the quality of the obtained DNA?
Response 7: Thank you for bringing this to our attention. We have added this information to the article (Page 4, lines 168-176):
Quality-filtered by fastp version 0.19.6 and merged by FLASH version 1.2.11 with the following criteria: (i) the 300 bp reads were truncated at any site receiving an average quality score of <20 over a 50 bp sliding window, and the truncated reads shorter than 50 bp were discarded, reads containing ambiguous characters were also discarded; (ii) only overlapping sequences longer than 10 bp were assembled according to their overlapped sequence. The maximum mismatch ratio of overlap region is 0.2. Reads that could not be assembled were discarded; (iii) Samples were distinguished according to the barcode and primers, and the sequence direction was adjusted, exact barcode matching, 2 nucleotide mismatch in primer matching.
Comments 8: It should be explained why the process of clustering OTUs at the 97% similarity level is important.
Response 8: Thank you for your comment. OTU (operational taxonomic units) is a taxonomic unit that is artificially labeled for ease of analysis in phylogenetics or population genetics studies. Sequences are usually divided into OTUs according to a 97% similarity threshold, and each OTU is usually considered as a microbial species. Sequences with less than 97% similarity are considered to belong to different species, and those with less than 95% similarity are considered to belong to different genera.
Comments 9: It should be explained how the network properties (average degree, average clustering coefficient, and density) are interpreted.
Response 9: Thank you for bringing this to our attention. We have added this information to the article (Page 4, lines 189-191):
Network images were generated using Cytoscape 3.9.1 and network properties (i.e. Average degree, Average clustering coefficient and Density) were calculated, which characterize the degree of densification and clustering of the network.
Comments 10: Results and discussions: Should it be explained more clearly why these differences in the relative abundance of the genera are significant, i.e., a higher or lower abundance of genera in one group compared to another?
Response 10: We sincerely appreciate the valuable comments. In this study, we analyzed the variability of baseline information and confounders (gender, age, smoking and alcohol consumption) between the two groups, and the results showed that there was no significant difference in the above indicators. Also, by reviewing the articles, we found that dietary factors have little influence on the composition of oral microbial communities (Wade, 2021). Therefore, we concluded that the different microbial composition in the two groups was due to heavy metal exposure.
Wade WG. Resilience of the oral microbiome. Periodontol 2000. 2021 Jun;86(1):113-122. doi: 10.1111/prd.12365.
Comments 11: What is the RANOSIM test, and how does it explain the distribution of fungal communities between groups?
Response 11: Thank you for your comment. We are very sorry for the spelling error, “RANOSIM” should be “ANOSIM”, the error has been corrected in the article (Page 7, line 261). ANOSIM test is a nonparametric test based on the permutation test and the rank sum test, which is used to test whether differences between groups are significantly larger than differences within groups. It is commonly utilized to test the similarity between High-dimensional data. PCoA analysis cannot calculate significant differences, therefore the ANOSM test is usually used in combination with PCoA analysis in microbiological analyses.
Comments 12: How might heavy metal exposure influence observed changes in the proportion of saprotrophs and pathotrophs between groups?
Response 12: Thank you for your feedback. In this study, we initially inferred that the changes in relative abundance were due to differences in the tolerance of fungi to heavy metals. However, the effects of heavy metals on oral fungi need to be further investigated, and no specific mechanism can be derived at present. We will continue to explore this area in the future. Thank you for your suggestion.
Comments 13: What is the significance of the fact that there are several exclusive OTUs in the exposure group?
Response 13: Thank you for your comment. We have added this information to the article (Page 9, lines 288-292):
The exclusive OTUs in the exposure group were greater than the shared OTUs, which suggested that heavy metal exposure may have disrupted the homeostasis of oral fungi, leading to the entry of more exogenous fungi (Sato, et al., 2021). We concluded that the species composition of the two groups was different.
Comments 14: What does each category of nodes represent in the analysis of ecological networks (for example, "connector" type nodes or "network hubs"? What impact do these categories have on the network structure and its stability?
Response 14: Thank you for your comment. Nodes were categorized into four classes based on within-module connectivity (Zi) and between-module connectivity (Pi). Peripherals represent unimportant nodes, which do not have high connectivity. Connectors represent nodes with high connectivity between two modules. Module hubs represent nodes with high connectivity within modules. Network hubs represent nodes with high connectivity throughout the network. Except for Peripherals, the other 3 types of nodes are keystone taxa that play an important role in the network and affect the stability of their ecosystems (Page 11, lines 333-336).
Nodes were categorized into four classes based on within-module connectivity (Zi) and between-module connectivity (Pi): peripherals (Zi ≤ 2.5, Pi ≤ 0.62), connectors (Zi ≤ 2.5, Pi ≥ 0.62), module hubs (Zi ≤ 2.5, Pi ≥ 0.62) and network hubs (Zi ≥ 2.5, Pi ≥ 0.62). Among them, connectors, module hubs and network hubs were defined as keystone taxa (Fig. 6A).
Comments 15: What is the significance of the fact that no key taxa were detected in the exposure group? Does this influence the interpretation of fungal network stability?
Response 15: Thank you for your comment. Keystone taxa are important species that have a significant impact on microbial community structure. It has been shown that keystone taxa play a particularly important role in the stability of their ecosystems. Therefore, the decrease in keystone taxa could lead to a decline in the stability of the fungal co-occurrence network, which is consistent with the results of the topological characteristics. We have added this information to the article (Page 2, lines 92-95):
Keystone taxa are important species that have a significant impact on microbial community structure, or even drive the covariation of the community. It has also been shown that keystone taxa play a particularly important role in the stability of their ecosystems (Amit and Bashan, 2023).
Comments 16: It should be more clearly explained what the saprotrophic mode means, what role it plays in the oral ecosystem, and why exposure to heavy metals appears to inhibit the growth of Saprotrophs.
Response 16: We sincerely appreciate the valuable comments. Currently, research on oral microbiology focuses mainly on oral bacteria, with relatively few fungi. Therefore, Relative research gap in this area. In this study, we speculated that the trend in relative abundance of fungi was caused by differences in the tolerance of fungi to heavy metals. Since molecular and cellular studies were not addressed in this paper, results on specific mechanisms could not be obtained. Similarly, the specific role of saprophytic fungi in the human oral cavity remains to be further investigated.
Comments 17: How the changes in the networks influence the ecological balance of the oral microbiome should be detailed.
Response 17: Thank you for your feedback. We have added this information to the article (Page 15, lines 443-452):
Ecological network analysis can be used to understand potential ecological associations among microbial communities and illustrate the effects of environmental heavy metal exposure on oral fungi in local populations. Compared to the exposure group, the complexity of the control group was mainly in total nodes, total links and average degree. Eight keystone taxa were identified in the control group, while no keystone taxa were identified in the exposure group. Keystone taxa in the control group belongs to Ascomycota and Basidiomycota at the phylum level, which are dominant phyla. The above results suggested that dominant phylum has a greater potential to become keystone taxa, and heavy metal exposure led to the loss of keystone taxa and imperiled the stability of the community.
Comments 18: What are the limitations of this study?
Response 18: Thank you for your comment.
First, the age composition of the population in our study was relatively homogeneous and did not involve infants, children and adolescents. The oral fungal composition and response to heavy metals may differ among different age groups, which need further study.
Secondly, the sequencing method used in this paper is amplicon sequencing, which has some limitations in functional annotation. Functional annotation should be further verified in combination with metagenomics and metabolomics subsequently.
